

# Divergent recovery trajectories of intertidal and subtidal coral communities highlight habitat-specific recovery dynamics following bleaching in an extreme macrotidal reef environment

P. Elias Speelman[1], Michael Parger[2] and Verena Schoepf[1,2]

[1] Institute for Biodiversity and Ecosystem Dynamics, Dept. of Freshwater and Marine Ecology, University of Amsterdam, Amsterdam, The Netherlands
[2] UWA Ocean Institute, The University of Western Australia, Perth, WA, Australia

Corresponding author
Verena Schoepf, v.schoepf@uva.nl

## ABSTRACT

Coral reefs face an uncertain future punctuated by recurring climate-induced disturbances. Understanding how reefs can recover from and reassemble after mass bleaching events is therefore important to predict their responses and persistence in a rapidly changing ocean. On naturally extreme reefs characterized by strong daily temperature variability, coral heat tolerance can vary significantly over small spatial gradients but it remains poorly understood how this impacts bleaching resilience and recovery dynamics, despite their importance as resilience hotspots and potential refugia. In the macrotidal Kimberley region in NW Australia, the 2016 global mass bleaching event had a strong habitat-specific impact on intertidal and subtidal coral communities at our study site: corals in the thermally variable intertidal bleached less severely and recovered within six months, while 68% of corals in the moderately variable subtidal died. We therefore conducted benthic surveys 3.5 years after the bleaching event to determine potential changes in benthic cover and coral community composition. In the subtidal, we documented substantial increases in algal cover and live coral cover had not fully recovered to pre-bleaching levels. Furthermore, the subtidal coral community shifted from being dominated by branching *Acropora* corals with a competitive life history strategy to opportunistic, weedy *Pocillopora* corals which likely has implications for the functioning and stress resilience of this novel coral community. In contrast, no shifts in algal and live coral cover or coral community composition occurred in the intertidal. These findings demonstrate that differences in coral heat tolerance across small spatial scales can have large consequences for bleaching resilience and that spatial patchiness in recovery trajectories and community reassembly after bleaching might be a common feature on thermally variable reefs. Our findings further confirm that reefs adapted to high daily temperature variability play a key role as resilience hotspots under current climate conditions, but their ability to do so may be limited under intensifying ocean warming.

## INTRODUCTION

Coral reefs are among the most productive and biodiverse ecosystems on the planet and provide income, food and resources to millions of people (*Moberg & Folke, 1999*; *Brander, Van Beukering & Cesar, 2007*; *Fisher et al., 2015*). However, they are threatened by a wide range of anthropogenic impacts ranging from climate change, ocean acidification and eutrophication to overfishing and invasive species (*Hoegh-Guldberg et al., 2007*; *Mumby & Steneck, 2008*). Rising global sea surface temperature (SST) in combination with marine heatwaves have led to widespread coral mass bleaching events that will increase in frequency and intensity as climate change intensifies (*Hughes et al., 2017*; *Hughes et al., 2018a*; *Frölicher, Fischer & Gruber, 2018*). Coral bleaching is a process where heat and/or light stress leads to the breakdown of the symbiosis between the coral and its endosymbiotic algae (family Symbiodiniaceae), resulting in a significant loss of algae from the coral tissue. As the coral host meets most of its energetic requirements from carbon and nutrients acquired autotrophically by the algal symbiont (*Muscatine, McCloskey & Marian, 1981*), bleaching leads to significant resource limitation as well as cytotoxic stress, and can result in the death of the coral colony if stress is severe or lasts for prolonged periods of time (*Oakley & Davy, 2018*). Coral mass bleaching events can therefore lead to coral mortality on regional to global scales and are one of the key threats to coral reefs today (*Eakin et al., 2010*; *Hughes et al., 2018b*; *Eakin, Sweatman & Brainard, 2019*).

As climate-induced disturbances increasingly impact coral reefs globally, understanding how reefs can recover from and reassemble after bleaching events is important to predict their responses and persistence in a rapidly changing ocean. Since mass bleaching can lead to widespread loss of live coral cover, it is particularly important to understand which factors and mechanisms allow reefs to recover *versus* those that drive regime shifts towards non-coral dominated states (*Graham, Nash & Kool, 2011*; *Graham et al., 2015*; *Arif et al., 2022*). For example, over the last decades, many coral reefs have shifted from coral to algal- or animal-dominated states (*e.g.*, soft corals, sponges or ascidians) due to eutrophication, loss of grazers, disease outbreak, cyclone damage or mass bleaching (*McManus & Polsenberg, 2004*; *Dudgeon et al., 2010*; *Graham et al., 2015*; *Bell, Micaroni & Strano, 2021*). These new communities often represent alternative stable states that are difficult to reverse (*Bellwood et al., 2004*), thus being of major concern for coral reef conservation and management. However, coral reefs have the potential to recover from catastrophic disturbances, though full recovery of coral assemblages generally takes from 10 to 15 years for the fastest growing species and far longer for the full complement of life histories and morphologies of older assemblages (*Gilmour et al., 2013*; *McClanahan & Muthiga, 2014*; *Glynn et al., 2015*; *Hughes et al., 2018a*).

In recent years, coral populations with naturally elevated heat resistance have been documented in environmentally extreme reef environments characterized by high temperature variability (*Palumbi et al., 2014*; *Schoepf et al., 2015*; *Safaie et al., 2018*). Since these locations represent resilience hotspots and potential climate change refuges, they play a critical role in facilitating future coral reef survival under rapid climate change as well as for marine spatial planning and conservation (*Camp et al., 2018*; *Burt et al., 2020*). Thermally

variable reefs have been increasingly studied to investigate the mechanisms underlying the enhanced heat tolerance of resident coral populations (*Barshis et al., 2013*; *Palumbi et al., 2014*; *Jung et al., 2021*; *Thomas et al., 2022*). However, it remains poorly understood how they respond to, and recover from, climate-induced disturbances (*Morikawa & Palumbi, 2019*; *Schoepf et al., 2020*; *Klepac & Barshis, 2020*), particularly over longer time scales such as years. Despite the enhanced heat tolerance of their resident coral populations, bleaching events have been documented on some thermally variable and extreme reefs as marine heatwaves increase in frequency and intensity across the globe (*Le Nohaïc et al., 2017*; *Morikawa & Palumbi, 2019*; *Klepac & Barshis, 2020*). Yet, we currently do not know whether these reefs recover from bleaching events in the same way as reefs in more typical, environmentally more benign environments which have informed most of our knowledge on coral reef recovery dynamics (*Graham, Nash & Kool, 2011*; *Graham et al., 2015*; *Arif et al., 2022*).

One such environmentally extreme reef location is the macrotidal Kimberley region in NW Australia which is characterized by the world's largest tropical tides yet nevertheless has abundant and highly diverse coral reefs (*Richards et al., 2015*). This extreme tidal regime exposes resident corals to strong currents and high turbidity, with shallow corals experiencing large daily temperature fluctuations (up to 8 °C) and regular aerial exposure at low tide that can last for several hours (*Dandan et al., 2015*; *Schoepf et al., 2015*) (Figs. 1E–1F). Thus, strong environmental gradients exist over small spatial scales that have resulted in enhanced heat tolerance of corals in the highly variable intertidal compared to conspecifics in the thermally less variable subtidal (*Schoepf et al., 2015*). Nevertheless, despite their ability to tolerate extreme environmental conditions, many coral reefs in the Kimberley region bleached extensively during the third documented global mass bleaching event in 2016 (*Le Nohaïc et al., 2017*; *Richards et al., 2019*; *Gilmour et al., 2019*). Interestingly, intertidal corals bleached less severely and recovered rapidly whereas the subtidal coral community suffered extensive loss of live coral cover (68%) six months after the bleaching event (*Schoepf et al., 2020*; *Jung et al., 2021*). However, it is currently unknown if the subtidal coral community was able to recover over the following years or whether shifts in coral community composition or toward non-coral dominated states have occurred. Furthermore, it is unknown whether the intertidal coral community was able to maintain its rapid return to pre-bleaching configuration in the long-term.

Here, we investigated how the intertidal and subtidal coral community at a well-studied reef in the inshore Kimberley region had recovered from the 2016 mass bleaching event 3.5 years later and compared these data to previous benthic surveys conducted at the same reef before and during the mass bleaching event (*Le Nohaïc et al., 2017*) and six months after bleaching (*Schoepf et al., 2020*; *Jung et al., 2021*). Given that widespread coral mortality had occurred in the subtidal but not intertidal, the aim of this study was to investigate habitat-specific recovery trajectories. Specifically, we asked the following research questions: (1) Did live coral cover in the subtidal recover to pre-bleaching levels over 3.5 years? (2) If not, did this result in changes in benthic cover indicative of a regime shift to non-coral dominated states? (3) Did either the subtidal or intertidal coral community experience a shift in coral community composition in response to the bleaching event? (4) If yes, did

<br />

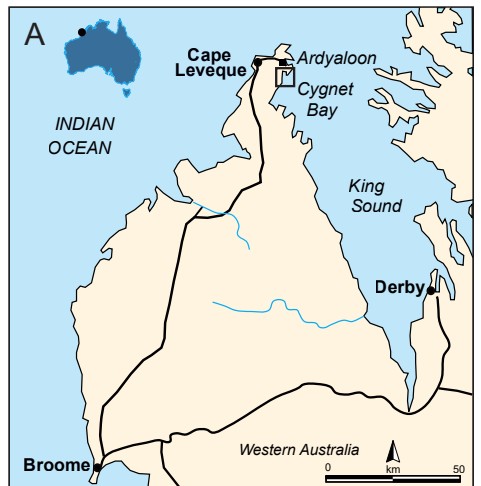
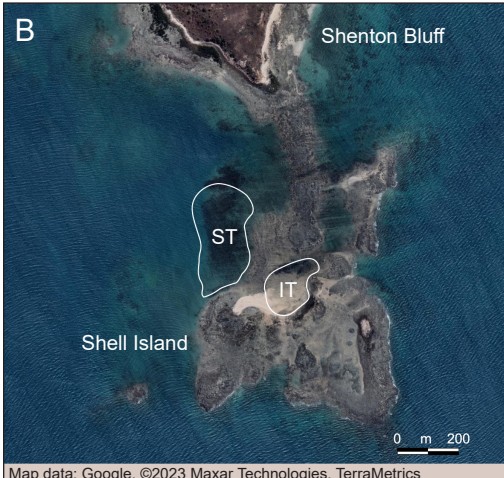
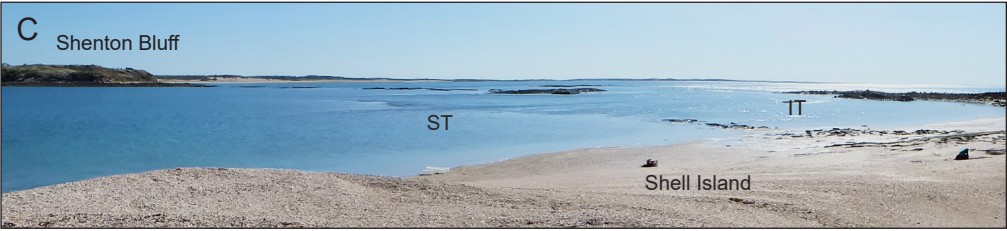
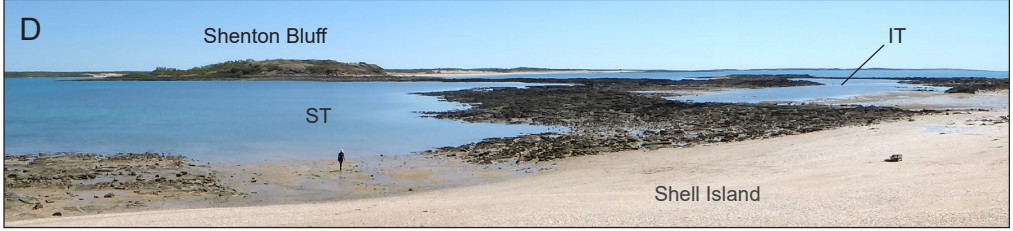
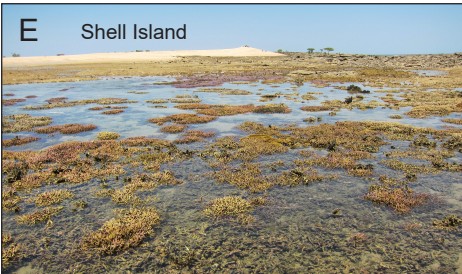
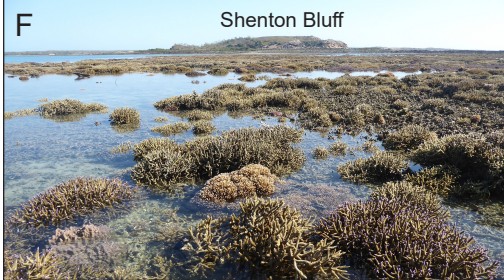

**Figure 1  Overview of study site.** (A) Dampier Peninsula and Cape Leveque in the southern Kimberley region, NW Australia, with black square showing the study site area near Cygnet Bay. Modified from Wikipedia (Cape Leveque Road —User: Summerdrought). (B) Close-up of the study area showing the intertidal (IT) and subtidal (ST) zone at Shell Island where surveys were conducted. Source: Google ©2023 Maxar Technologies, TerraMetrics. (C) Study area at high tide. (D) Study area during outgoing tide, showing emerging intertidal pool and subtidal zone. Corals exposed at low tide in the (E) intertidal pool and (F) subtidal zone. Photo credit (C–F): V. Schoepf.

this reshuffling also result in a functional community shift due to changes in dominant life history strategies, coral morphologies or community bleaching resistance? Our study provides an important contribution to the understanding of the short-term recovery dynamics (several years) of coral reefs (see also *Tebbett, Morais & Bellwood, 2022*) which is increasingly important as recovery intervals between consecutive bleaching events are becoming shorter under rapid climate change (*Hughes et al., 2018a*).

## MATERIALS & METHODS

### Study site

The study site was located at Shell Island (Shenton Bluff), Cygnet Bay, in the macrotidal Kimberley region in North Western Australia (Fig. 1). This region has the highest tropical tidal range (semidiurnal, up to 12 m), with Shell Island experiencing a maximum tidal range of ~8 m. The study site has been described in detail previously (*Dandan et al., 2015*; *Schoepf et al., 2015*; *Schoepf et al., 2020*). Briefly, during spring low tides, an intertidal pool is formed, creating an isolated habitat (Figs. 1B–1E). This intertidal pool (16°28′45.8″S, 123°2′41.3″E) is about 200 by 100 m wide and retains a minimum depth of 20 to 30 cm during low tide (*Schoepf et al., 2020*). When depth data are averaged over long time periods, this zone has an average depth of ~3 m (*Schoepf et al., 2020*). Even though the pool always retains a low water level at low tide, many coral colonies are regularly exposed to air for a few hours (Fig. 1E). This pool has a high variability in daily temperatures of up to 8 °C (*Schoepf et al., 2015*; *Schoepf et al., 2020*). The subtidal zone (16°28′46.8″S, 123°2′36.6″E; within 200–300 m of the intertidal) represents a more moderate thermal environment with an average depth of 4 m and only up to 4 °C daily temperature range (Figs. 1B–1D and 1F). Both zones experience high turbidity due to tidal currents, although light levels can reach up to up to 2,400 $\mu$mol m$^{-2}$ s$^{-1}$ during low tide (*Dandan et al., 2015*; *Schoepf et al., 2020*).

### Environmental monitoring

In both intertidal and subtidal zones, water temperature was recorded continuously from 1 September 2015 until 23 July 2018 and from 15 December 2018 until 1 October 2019. Temperature was recorded at the depth of the corals (approx. 20 cm above the substrate) using HOBO U22 v2 temperature loggers ($\pm$0.2 °C) every 15 min. Cumulative heat stress was calculated following *Schoepf et al. (2020)* using the local Monthly Maximum Mean (MMM) temperature of 30.827 °C from the U.S. National Oceanic and Atmospheric Administration's (NOAA) 5-km virtual station North Western Australia (version 2). Weekly temperatures were calculated and all positive anomalies exceeding the MMM over a period of 12 weeks accumulated, resulting in a metric termed "w > MMM" (*Schoepf et al., 2020*). This metric can be compared to NOAA's Degree Heating Weeks (DHW), with the difference that NOAA's DHW method only accumulates positive anomalies exceeding the local MMM by 1 °C (*Liu et al., 2014*).

We further calculated three metrics that characterize the distribution of the temperature data because historic temperature regimes strongly influence coral susceptibility to bleaching (*McClanahan et al., 2007*; *Safaie et al., 2018*). To predict bleaching prevalence, we used the daily temperature range (DTR) because it is one of the most pertinent to highlight

such high-frequency temperature variability mitigates bleaching risk (*Safaie et al., 2018*). The DTR was previously reported for our study site only for the period 2011–2012 (*Schoepf et al., 2015*), so here we extend these data for the period 2015–2019 and calculated a separate DTR for both intertidal and subtidal by subtracting the minimum temperature from the maximum for every day. In addition, we calculated the skewness and kurtosis of the DTR for each zone to describe the DTR distribution shape. Skewness is a measure of the asymmetry of a distribution while kurtosis describes its flatness or "peakiness" (*Groeneveld & Meeden, 1984*). Positively skewed temperature distributions have been associated with site-specific increased thermal tolerance (*Baker et al., 2013*; *Zinke et al., 2018*), while reef areas with low kurtosis have been linked to reduced bleaching (*Ateweberhan & McClanahan, 2010*).

## Benthic surveys

The El Niño event in 2015–2016 caused a global marine heatwave that instigated the most extensive and severe global bleaching event documented to date (*Eakin, Sweatman & Brainard, 2019*). One of the areas affected by the bleaching event in the austral summer (April 2016) was the inshore Kimberley region (*Le Nohaïc et al., 2017*; *Gilmour et al., 2019*; *Schoepf et al., 2020*), though not every reef was affected (*Richards et al., 2019*). Reef-wide coral health surveys were conducted at low tide at Shell Island 3.5 years after peak bleaching (1–3 October 2019) using intertidal walking. The same methods were followed as those already used by *Le Nohaïc et al. (2017)* and *Schoepf et al. (2020)* to assess immediate impacts and recovery after six months at the same study site, respectively. These surveys were then compared to the previously published data collected prior to bleaching (January 2016), during peak-bleaching (April 2016) and after six months of recovery (October 2016) (*Le Nohaïc et al., 2017*; *Schoepf et al., 2020*). Six 15 meter transects were randomly deployed in both the intertidal and in the subtidal zone, as for previous time points with the exception of January 2016 when seven transects were used. High-resolution photos of a 50 by 50 cm quadrat were taken every 1 m (every 0.5 to 1 m at previous time points).

The benthic cover and coral community composition of the photo-quadrats was analyzed using PhotoQuad (*Trygonis & Sini, 2012*). The benthic cover was divided into the following categories (Fig. S1): live hard coral, soft coral, algae, substrate, recently deceased coral and "unknown/other" (anything that did not fit in the aforementioned categories or was unclear). The category substrate was used for all types of abiotic benthic cover such as rubble, sand or rocks. Corals were categorized as recently deceased if they were presumed to have died in the last few months, as indicated by a general coral shape still being recognizable. If a coral had already disintegrated into rubble, it was considered dead for a longer period of time and thus categorized as substrate. The live hard corals were identified to genus level, if possible. However, due to difficult conditions associated with the large tidal range, high turbidity and reflections in the waterline, the quality of the images was sometimes not sufficient to allow for identification to genus level. In that case, only the morphology was recorded.

## Indicators of reef function

We assessed three characteristics that could indicate potential shifts in reef function due to changes in coral community composition (*e.g.*, *McWilliam et al., 2020*), particularly in

the context of community bleaching resilience: coral morphology, life history strategies and the corallite integration index score (*Loya et al., 2001*; *Darling et al., 2012*; *Swain et al., 2018*). Coral morphology was recorded for each colony in the photo-quadrats, whereas the other two characteristics were attributed based on a colony belonging to a certain genus.

### Coral morphology

Coral morphology has often been associated with differential bleaching resistance, with branching and plating corals typically being more sensitive than massive or mounding species (*Marshall & Baird, 2000*; *Loya et al., 2001*). Furthermore, coral morphology has functional relevance because some growth forms such as branching or plating taxa contribute significantly more to the complexity of reef topography than *e.g.*, encrusting taxa (*e.g.*, *Alvarez-Filip et al., 2013*; *Husband, Perry & Lange, 2022*). We therefore recorded coral morphology for each colony using the following categories: massive, encrusting, branching, plating or solitary/free-living.

### Life history strategy (LHS)

Reef function and resistance to environmental change is strongly influenced by the species-specific morphological and physiological attributes that determine coral life history strategies (*Darling et al., 2012*; *Alvarez-Filip et al., 2013*; *McWilliam et al., 2020*). *Darling et al. (2012)* assigned four LHS to the most common species in the Indo-Pacific and the Atlantic: competitive, weedy, generalist and stress-tolerant strategies. Since this identification was based on species level, yet corals were only identified to genus level in this study, the corals recorded in the surveys were assigned a LHS using the following steps (Table S1). Generally, only species occurring in the Indo-Pacific were considered, and their occurrence had to be confirmed or strongly predicted for the Kimberley Coast, north-west Australia (ER091), as per Corals of the World, version 0.01 (*Veron et al., 2023*).

When a genus consisted of only a single species or when there were multiple species, but all species shared the same LHS, this LHS was assigned to the genus (*e.g.*, competitive LHS for *Acropora*). This was the case for 12 out of 26 recorded coral genera (Table S1). If there were multiple species with different LHS within a genus, the morphological differences between species were assessed and the possibility for species identification was evaluated. This was only possible for the genus *Pocillopora* (weedy or competitive). Although this genus consists of several species complexes (*Schmidt-Roach et al., 2013*; *Schmidt-Roach et al., 2014*), the dominant species recorded in the surveys was identified as *Pocillopora acuta* based on macro-morphology (Sebastian Schmidt-Roach, pers. comm., 2023) (Fig. S2). *Pocillopora acuta* is not listed in *Darling et al. (2012)* because it used to be synonymous with *P. damicornis* (type *β*) and we therefore assigned it the weedy LHS listed for that species. For five genera, species ID to differentiate between different LHS was not possible and they were therefore excluded from the LHS analysis (*i.e.*, *Goniastraea*, *Montipora*, *Pavona*, *Porites*, *Turbinaria*). Finally, not all genera recorded in this study were listed in *Darling et al. (2012)*; therefore, no LHS could be assigned to eight genera (*Coeloseris*, *Ctenactis*, *Euphyllia*, *Goniopora*, *Herpolitha*, *Leptoseris*, *Millepora* and *Trachyphyllia*) and they were therefore also excluded from the LHS analysis (Table S1).

### Corallite integration index score (CIIS)

Physiological integration of coral colonies has been shown to correlate with bleaching resistance because coral colonial integration and coordination improves responses to injury, predation, disease, and stress (*Swain et al., 2018*). We therefore assigned the Corallite Integration Index Score (CIIS) developed by *Swain et al. (2018)* to each coral genus, where higher scores are linked to a reduced bleaching response. As for the LHS, the CIIS was originally assigned to individual species. We therefore calculated genus-level CIIS by averaging all species scores as assigned by *Swain et al. (2018)* to each genus. Since this resulted in a wide range of genus-level scores, they were divided into two groups to facilitate multivariate analysis. Genera with an average CIIS of >2 were classified as "high CIIS" whereas those with a CIIS of <2 were classified as "low CIIS".

## Statistical analysis

To test for differences in daily temperature range between the two zones, a one-tailed student's $t$-test was performed in R. Permutational multivariate analysis of variance (PERMANOVA) was utilized to analyze the benthic survey data. Prior to analysis, all count data from the surveys was converted to percent abundance per quadrat and then square root transformed to reduce the influence of extremely high abundance of a single genus (*i.e., Acropora*). Five separate two-way PERMANOVAs were then conducted on benthic cover, coral community composition, growth form, life history strategy and corallite integration index score to test for differences between the two zones (two levels: intertidal, subtidal) and the survey timepoints (either two or four levels; two levels: pre-bleaching in January 2016, 3.5 year recovery in October 2019; four levels: pre-bleaching in January 2016, peak-bleaching in April 2016, six-month recovery in October 2016, 3.5 year recovery in October 2019). The Bray-Curtis similarity index was used with 999 permutations. *Post hoc* pairwise comparisons were calculated with *p*-value adjustments according to the Holm method. The transects served as replicates. Before running the PERMANOVAs, the PERMDISP function was run to test for homogeneity of multivariate dispersions, and the majority of PERMANOVAs (seven out of 10) fulfilled this assumption (Table S2). However, a non-significant test result is not strictly necessary as the PERMANOVA routine is relatively robust to heterogeneity of multivariate dispersions, particularly with large sample sizes and a balanced design (*Anderson, Gorley & Clarke, 2008*; *Anderson & Walsh, 2013*). Principal component analysis was used to visualize the data. All multivariate statistical analyses were performed in R with the packages 'vegan' and 'pairwiseadonis' (*Dixon, 2003*; *Martinez Arbizu, 2020*).

## RESULTS

### Environmental monitoring

Average temperatures from 1 September 2015 until 1 October 2019 were very similar in both environments with 28.60 °C ($\pm$2.58 °C, SD, $n = 123,729$) in the intertidal and 28.66 °C ($\pm$2.50 °C, SD, $n = 128,886$) in the subtidal (Fig. 2). In contrast, the maximum daily average temperature recorded in the intertidal was much higher than in the subtidal with 38.09 °C and 33.84 °C, respectively. The daily temperature range (DTR) was significantly higher in

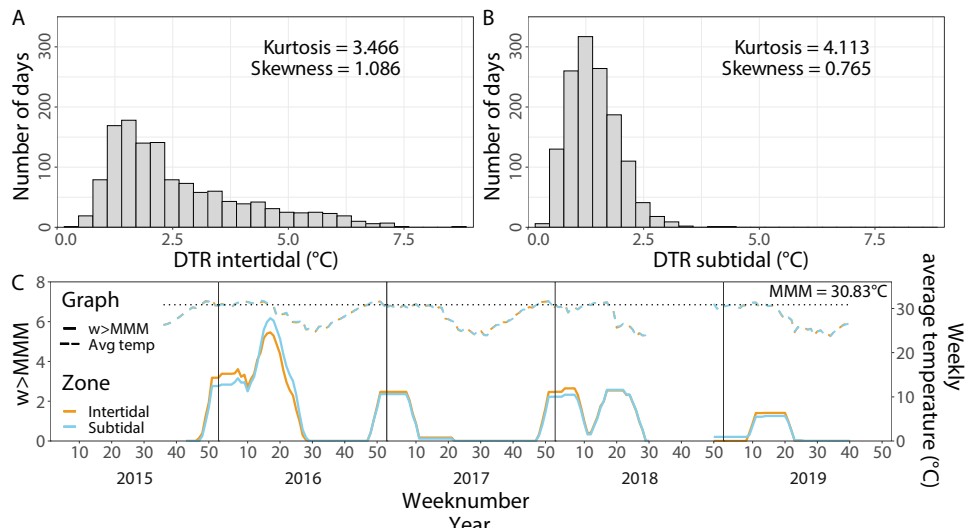

**Figure 2 _In situ_ seawater temperatures from September 2015 to October 2019.** Histogram of the daily temperature range (DTR) in (A) the intertidal and (B) the subtidal zone. (C) Weekly average temperatures (dashed lines) and cumulative heat stress (w >MMM, solid lines) over the period of September 2015 until October 2019 in the intertidal (orange) and subtidal (blue). The dotted line indicates the local maximum monthly mean (MMM) temperature.

the intertidal compared to the subtidal ($t = 27.182$, $df = 1,611.2$, $p < 2.2 \times 10^{-16}$), with a mean DTR of 2.6 °C ($\pm 1.49$ °C, SD, $n = 1,057$) and 1.4 °C ($\pm 0.54$ °C, SD, $n = 1,055$), respectively. Maximum DTR was 8.5 °C in the intertidal but only 4.5 °C in the subtidal (Figs. 2A & 2B). DTR kurtosis was lower in the intertidal (3.466) than in the subtidal (4.113) whereas the DTR skewness was higher in the intertidal (1.086) than in the subtidal (0.765) (Figs. 2A & 2B).

Contrary to the DTR, weekly average temperatures and cumulative heat stress were very similar across both environments (Fig. 2C). Heat stress during the late Austral summer/early autumn of 2016 reached around 5.5 w >MMM in both the intertidal and subtidal environments. A similar level of heat stress was not reached in the following three years although maximum w >MMM values of 2.5, 2.6 and 1.4 occurred in the late Austral summer and early autumn of 2017, 2018 and 2019, respectively.

## Benthic cover

Benthic cover at the study sites changed significantly over time (all four time points included), but this was dependent on habitat; in addition, the individual effects of time and zone were also significant (Figs. 3A & 3B, 4A, Table 1). In the intertidal, hard coral cover only decreased by 3% six months after peak bleaching compared to pre-bleaching (25%) but increased to 38% after 3.5 years of recovery. Algal cover stayed relatively constant (*i.e.,* 6–8% cover) throughout both recovery time points. In contrast, in the subtidal, hard coral cover decreased from 36% to 13% after the 2016 bleaching event, but increased to 25% after 3.5 years of recovery. Algal cover stayed low six months after the bleaching event (1–2% cover), but increased to 15% after 3.5 years of recovery. A separate PERMANOVA

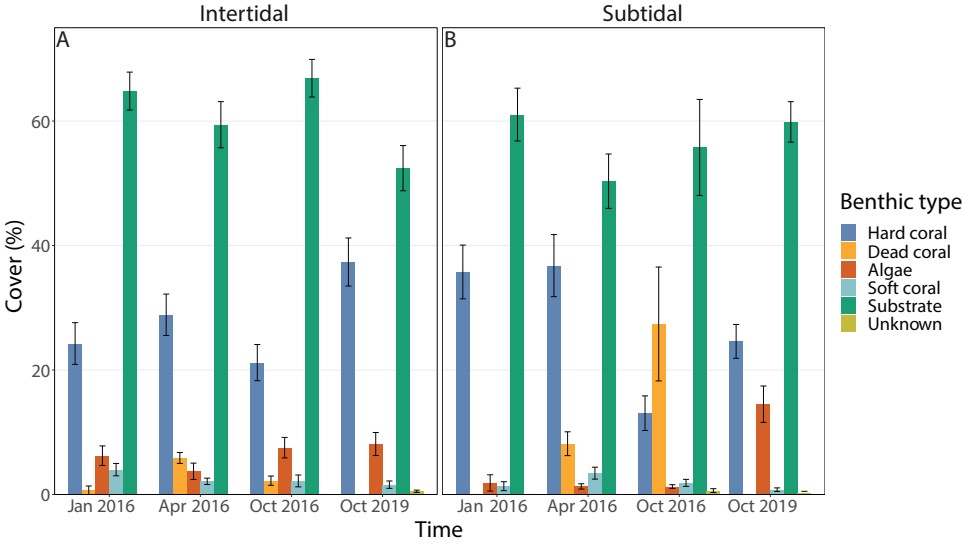

**Figure 3** **Percentage cover of all benthic categories across all four time points in the (A) intertidal and (B) subtidal zone.** Included are hard coral, soft coral, dead coral, algae, substrate (consists of all types of abiotic substrate types such as rock, rubble and sand) and unknown/other. Shown is mean ± 1SE.

comparing benthic cover only across two time points, *i.e.,* pre-bleaching and 3.5 years of recovery, revealed again a significant interactive effect of time and habitat as significant changes in benthic cover occurred only in the subtidal (Figs. 3A & 3B, 4A, Table S3). In contrast, intertidal benthic cover 3.5 years after the bleaching event was not significantly different from the pre-bleaching time point.

## Coral community composition

Coral community composition changed significantly over time (all four time points included) but similar to benthic cover this was also dependent on zone; the individual effects of time and zone were also significant (Figs. 4B, 5A & 5B, Table 1). In the intertidal, there were no significant shifts in coral community composition which was dominated by branching *Acropora* at all four time points (62% to 76% cover). The second most abundant genus in the intertidal was massive *Porites* (~10% cover). In contrast, there was a significant change in the subtidal coral community composition over time as it shifted from being dominated by branching *Acropora* only in 2016 (80% to 85% cover) to being dominated by both *Acropora* and *Pocillopora* after 3.5 years of recovery (50% and 42% cover, respectively) (Figs. 4B and 5B). These zone-specific changes in coral community composition over time were also confirmed in a separate PERMANOVA comparing the 3.5 year recovery time point to the pre-bleaching time point only: while intertidal coral community composition did not differ significantly between these two time points, this was not the case for the subtidal community composition (Fig. 4B, Table S3).

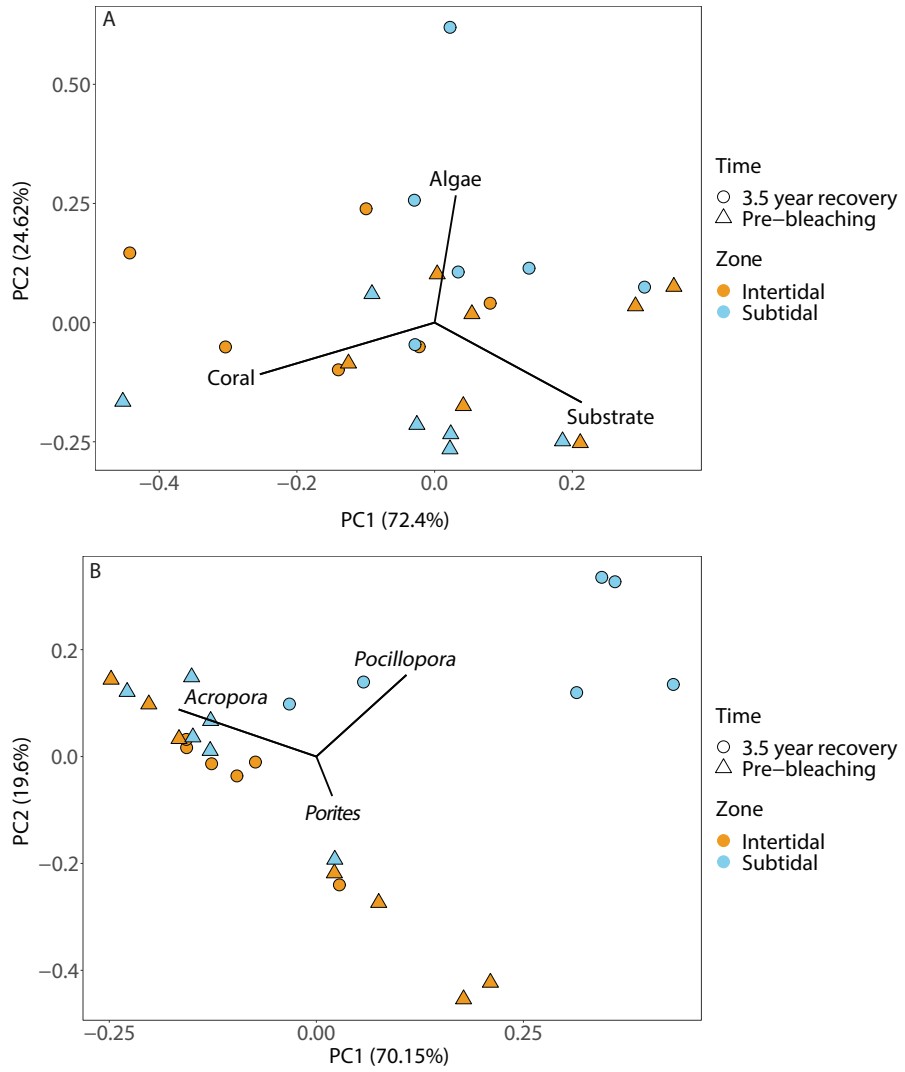

**Figure 4** **Principal component analysis showing (A) benthic cover and (B) coral community composition after 3.5 years of recovery (October 2019) and at the pre-bleaching time point (January 2016) for both intertidal and subtidal zone.** Vectors represent the benthic cover categories and coral genera that had the greatest influence on overall benthic cover and coral community composition, respectively.

## Indicators of reef function
### *Coral morphology*

The dominant coral morphology did not change over time but differed significantly between intertidal and subtidal zones (Fig. S3, Table 1, Table S3). Both zones were strongly dominated by branching coral colonies such as *Acropora* and *Pocillopora* but slightly higher percentages were observed in the subtidal (81% to 90%) compared to the intertidal (72% to 83%). In addition, 11% to 21% of coral colonies in the intertidal were massive coral colonies, whereas this was only 4% to 9% in the subtidal. Within the intertidal, massive

**Table 1  Results from Permanova analyses testing for the effect of time (all four time points) and zone (intertidal vs subtidal) on benthic cover, coral community composition, coral morphology, life history strategy and corallite integration index score (CIIS).** See Table S3 for separate analyses comparing only 3.5 year recovery time point (Oct 2019) to the pre-bleaching time point (Jan 2016). Significant *p*-values are highlighted in bold.

| | Factors | Df | SumsOfSqs | MeanSqs | F.Model | R2 | Pr (>F) |
|---|---|---|---|---|---|---|---|
| **Benthic cover** | Time | 3 | 0.408 | 0.136 | 7.403 | 0.276 | **0.001** |
| (4 time points) | Zone | 1 | 0.066 | 0.066 | 3.602 | 0.045 | **0.018** |
| | Time:Zone | 3 | 0.250 | 0.083 | 4.543 | 0.169 | **0.001** |
| | Residuals | 41 | 0.753 | 0.018 | | 0.510 | |
| | Total | 48 | 1.478 | | | 1.000 | |
| **Coral comm.** | Time | 3 | 1.084 | 0.361 | 4.081 | 0.173 | **0.001** |
| **composition** | Zone | 1 | 0.465 | 0.465 | 5.246 | 0.074 | **0.001** |
| (4 time points) | Time:Zone | 3 | 1.098 | 0.366 | 4.134 | 0.175 | **0.001** |
| | Residuals | 41 | 3.631 | 0.089 | | 0.578 | |
| | Total | 48 | 6.278 | | | 1.000 | |
| **Coral** | Time | 3 | 0.124 | 0.041 | 0.933 | 0.055 | 0.461 |
| **morphology** | Zone | 1 | 0.241 | 0.241 | 5.456 | 0.107 | **0.010** |
| (4 time points) | Time:Zone | 3 | 0.071 | 0.024 | 0.539 | 0.032 | 0.774 |
| | Residuals | 41 | 1.810 | 0.044 | | 0.806 | |
| | Total | 48 | 2.246 | | | 1.000 | |
| **Life history** | Time | 3 | 0.355 | 0.118 | 5.015 | 0.175 | **0.002** |
| **strategy** | Zone | 1 | 0.328 | 0.328 | 13.900 | 0.162 | **0.001** |
| (4 time points) | Time:Zone | 3 | 0.379 | 0.126 | 5.350 | 0.187 | **0.002** |
| | Residuals | 41 | 0.968 | 0.024 | | 0.477 | |
| | Total | 48 | 2.030 | | | 1.000 | |
| **Corallite** | Time | 3 | 0.283 | 0.095 | 5.788 | 0.185 | **0.003** |
| **integration** | Zone | 1 | 0.433 | 0.433 | 26.522 | 0.283 | **0.001** |
| **index score** | Time:Zone | 3 | 0.146 | 0.049 | 2.991 | 0.096 | **0.045** |
| (4 time points) | Residuals | 41 | 0.669 | 0.016 | | 0.437 | |
| | Total | 48 | 1.532 | | | 1.000 | |

coral colonies made up only 11% of the coral community 3.5 years after the mass bleaching event whereas this ranged from 17% to 21% for the three time points in 2016.

### Life history strategy (LHS)

The dominant life history strategy changed significantly over time but this was also dependent on zone; in addition, both the effects of time and zone were significant on their own (Fig. 6, Table 1). The coral community in both zones was generally dominated by genera with a competitive LHS (mostly branching *Acropora*). However, in the intertidal, this was the case across all four time points (93% to 98%), whereas in the subtidal this dropped from 94% to 96% during the three time points in 2016 to only 53% after 3.5 years of recovery. This drop coincided with a strong increase in corals with a weedy LHS (mostly *Pocillopora*), which made up 2% to 5% of the subtidal coral community in 2016 but increased to 44% after 3.5 years of recovery (Fig. S2). In contrast, the percentage cover of corals with a weedy LHS remained low and stable in the intertidal (<2%) across all four time points. The PERMANOVA comparing the 3.5 year recovery time point only to

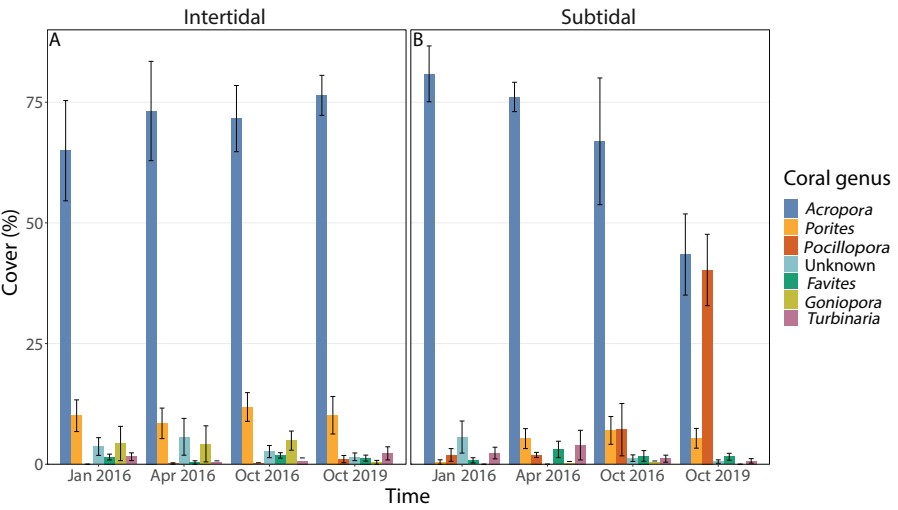

**Figure 5  Percentage cover of the seven most abundant coral genera across all four time points in the (A) intertidal and (B) subtidal zone.** Shown is mean ± 1SE.

pre-bleaching further confirmed the significant interaction of time and zone, and showed that the subtidal coral community had significantly different LHS after 3.5 years of recovery compared to both pre-bleaching and the intertidal community at either time point (Fig. 6C, Table S3).

### *Corallite integration index score (CIIS)*

The corallite integration index score changed significantly over time but this was also dependent on zone (Fig. S4, Table 1, Table S3). In addition, the effects of time and zone were significant on their own (Table 1, Table S3). In general, both intertidal and subtidal zones were dominated by coral genera with high CIIS (*e.g.*, *Acropora*) at almost all time points. However, the intertidal coral community showed no significant change over time (85% to 91% at all four time points), whereas the subtidal shifted from high CIIS dominance in 2016 (76% to 78%) to low CIIS dominance (55%) after 3.5 years of recovery. This coincided with the corresponding shift in dominance from *Acropora* (high CIIS, competitive LHS) to *Pocillopora* (low CIIS, weedy LHS).

## DISCUSSION

As the frequency and intensity of marine heatwaves increasingly threatens coral reefs (*Frölicher, Fischer & Gruber, 2018*), it has become ever more important to identify naturally heat-resistant coral populations that are capable of coping with intensifying heat stress. Such populations exist, for example, in thermally extreme reef environments, such as reefs with high-frequency temperature variability, and these populations have been shown to bleach less severely during marine heatwaves and exhibit high bleaching resilience (*e.g.*, *Safaie et al., 2018*; *Schoepf et al., 2020*). However, it remains poorly understood how corals from thermally variable reef habitats recover from bleaching over longer time scales (~years). Here, we show that recovery in the moderately variable, subtidal environment was

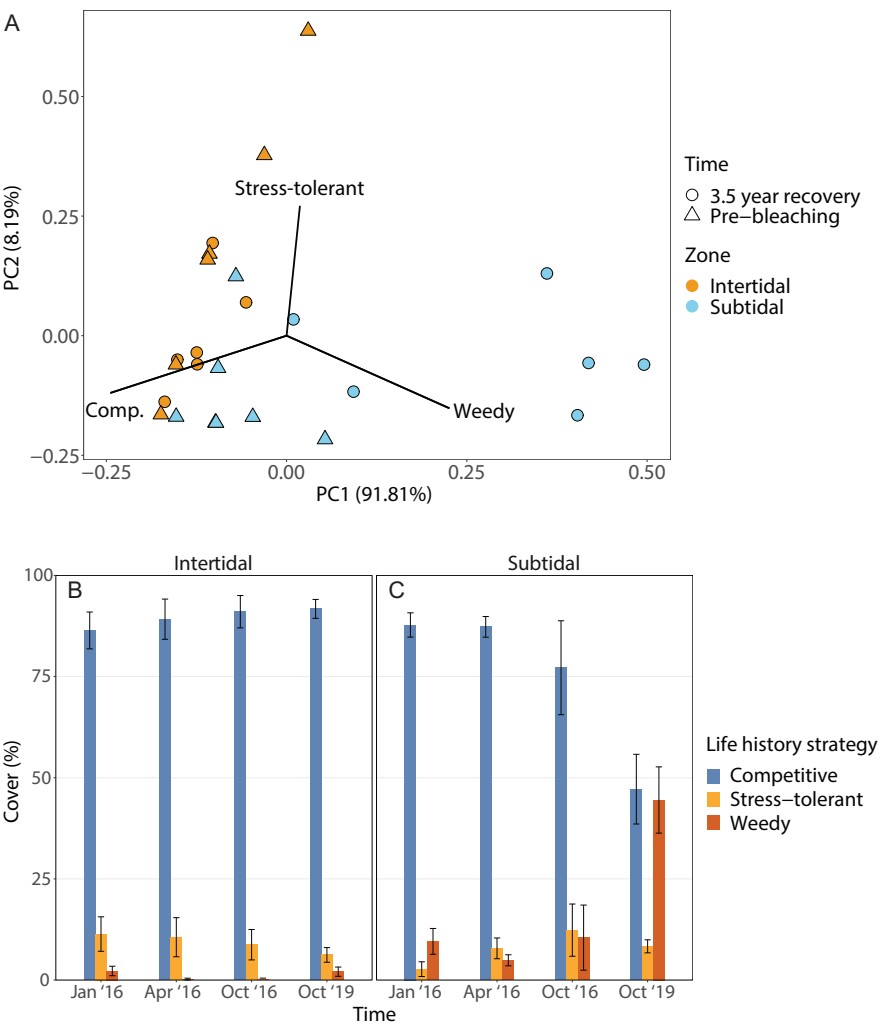

**Figure 6 Life history strategies.** (A) Principal component analysis showing life history strategy after 3.5 years of recovery (October 2019) and at the pre-bleaching time point (January 2016) for both intertidal and subtidal zone. Vectors represent the life history strategies that had the greatest influence. (B, C) Percentage cover of coral genera with a certain life history strategy across all four time point in the (B) intertidal and (C) subtidal zone. Shown is mean ± SE.

incomplete 3.5 years after mass bleaching and extensive coral mortality, whereas the coral community in the highly variable intertidal had already returned to their pre-bleaching configuration within six months. Furthermore, the subtidal coral community showed a shift in community composition that did not occur in the intertidal over the same time period, indicating habitat-specific divergence in recovery trajectories.

## Increases in algal cover and incomplete coral recovery in the subtidal zone

Following extensive coral mortality after the 2016 mass bleaching (68% loss of live coral cover in the subtidal) (*Schoepf et al., 2020*), significant changes in benthic cover had

occurred in the subtidal after 3.5 years, with the coral community showing only partial or incomplete recovery (Fig. 4). When mass bleaching leads to severe reductions in live coral cover and recovery is impaired, this can lead to 'regime' or 'phase' shift towards a system dominated by taxa other than scleractinian corals, such as macro-algae or soft coral (*McManus & Polsenberg, 2004*; *Bell, Micaroni & Strano, 2021*). In the subtidal, algal cover increased from 2% prior to the bleaching event to 15% 3.5 years later whereas it remained low (4–8% cover) and stable in the intertidal throughout this time period (Fig. 3). Nevertheless, this substantial increase in subtidal macroalgal cover does not represent a phase shift towards an algae-dominated state because they are defined as algae having 50% absolute cover or algal cover exceeding live coral cover—neither of which was the case here (*Norström et al., 2009*; *Bruno et al., 2009*). It remains to be seen whether algal cover will further increase or decrease over the coming years as recovery continues.

It is technically possible that the increase in macroalgal cover in the subtidal represents heterogeneity in benthic cover rather than an actual increase over time since we did not use permanent transects. However, we consider this unlikely given that random transects were used at all time points and in both reef zones and covered the same approximate area, yet macroalgal cover did not fluctuate by more than 1–4% in either zone across the other time points. It is further unlikely that the observed changes in benthic cover occurred due to other heat-related environmental changes. The temperature logger data show that no further prolonged heat stress event occurred in the 3.5 years after the 2016 bleaching events, although some heat stress (maximum w > MMM values of up 2.6) also occurred in late summer/autumn 2017–2019 (Fig. 2). We are also not aware of any reports of major bleaching during this time period. Furthermore, weekly average temperatures were highly similar in both reef zones. Therefore, changes in benthic cover and the increase in subtidal macroalgal cover, in particular, highlight the differential impact that the 2016 bleaching event had on the two different reef zones.

Recovery trajectories differed markedly across the two reef zones. Recovery of the subtidal coral community was incomplete 3.5 years after the mass bleaching event because live coral cover had not fully recovered to pre-bleaching levels and a significant shift in community composition had occurred (Figs. 4 and 5). This was in stark contrast to the intertidal coral community which showed full recovery and a return to its pre-bleaching configuration already within six months after the bleaching event (*Schoepf et al., 2020*). Such habitat-specific divergence in recovery trajectories across small spatial scales (a few hundred meters) is remarkable but has also been documented in other locations and highlights the context-dependent nature of coral recovery (*Golbuu et al., 2007*; *Tebbett, Morais & Bellwood, 2022*; *Thomas et al., 2022*). High-frequency temperature variability, in particular, has been shown to enhance coral heat tolerance and reduce bleaching risk (*Palumbi et al., 2014*; *Schoepf et al., 2015*; *Safaie et al., 2018*) because frequent exposure to stressfully high temperatures can promote acclimatization and adaptation (*e.g.*, *Rivest, Comeau & Cornwall, 2017*). While weekly average temperatures were very similar in both reef zones, the daily temperature range (DTR) was significantly higher in the intertidal than subtidal (up to 8.5 °C *vs* up to 4.5 °C, respectively), with maximum short-term temperatures reaching 38.09 °C and 33.84 °C, respectively (Fig. 2). Positively skewed temperature

distributions, such as observed here for the intertidal DTR, have been associated with stress-tolerant corals (and lower coral cover) in Western Australia (*Zinke et al., 2018*), while reef areas with low kurtosis (intertidal < subtidal) have been linked to higher heat tolerance (*e.g., Ateweberhan & McClanahan, 2010*). Therefore, strong differences in daily temperature range across small spatial scales have not only resulted in differential heat tolerance of corals at our study site (*Schoepf et al., 2015*; *Le Nohaïc et al., 2017*) but also remarkable differences in bleaching resilience and recovery potential (*Le Nohaïc et al., 2017*; *Schoepf et al., 2020*).

Interestingly, there was also a large increase in live coral cover (38% cover) in the intertidal 3.5 years after the bleaching event compared to pre-bleaching (25% cover). Although the reasons for this trend are unknown, significant increases in live coral cover despite repeated disturbances have also been documented for other reefs, including Moorea (*Holbrook et al., 2018*) and another remote reef system in Western Australia, the Rowley Shoals (*Gilmour et al., 2019*). Since we are not aware of any other major disturbances in 2015 or shortly before the bleaching events in April 2016, it is likely that this increase in coral cover reflects natural year-to-year variation in coral cover and/or the fact that transects were not permanent.

In the subtidal, extensive mortality due to the bleaching event resulted in only 13% live coral cover six months after bleaching, compared to 36% live coral cover prior to the bleaching event. Therefore, the 25% live coral cover measured after 3.5 years of recovery indicates substantial, yet at present incomplete recovery. This is not surprising given that full recovery of coral assemblages post bleaching tends to take at least 10 to 15 years (*Gilmour et al., 2013*; *Hughes et al., 2018a*; *Gouezo et al., 2019*). However, this can also depend on whether there is a shift between species, and some coral communities have shown rapid recovery after the 2016 mass bleaching event (within 4 years) (*e.g., Nakamura et al., 2022*). The extreme seascape of the macrotidal Kimberley region represents significant barriers to larval dispersal (*Underwood et al., 2020*). However, we nevertheless consider it likely that further recovery of live coral cover will take place over the coming years for several reasons, at least in the absence of further disturbances. First, many shallow coral reefs in the Kimberley are dominated by broad-cast spawning corals such as *Acropora* which show greater connectivity than brooders (*Underwood et al., 2020*). Second, the completed recovery and close proximity (200–300 m) of the intertidal coral community will likely enhance local recruitment and potentially supply more heat-resistant genotypes. Finally, the Kimberley region is a remote, near-pristine marine environment (*Richards et al., 2019*), thus the absence of local stressors in combination with low initial macroalgal cover and high branching coral cover is likely to promote coral reef recovery (*Graham et al., 2015*; *Donovan et al., 2021*; *Arif et al., 2022*).

## Shifts in coral community composition in the subtidal

The incomplete recovery of live coral cover in the subtidal was driven by high recruitment of *Pocillopora* corals which resulted in a strong shift in community composition as *Pocillopora* cover increased from 2% prior to the bleaching event to 44% cover 3.5 years post bleaching (mostly small colonies) (Figs. 4B and 5). As a consequence, the coral

community became dominated by *Pocillopora* (followed closely by *Acropora* with 40% cover, Fig. S2), even though *Acropora* made up 81% of live coral cover prior to bleaching. As brooders, *Pocillopora* corals can have significant advantages over broadcast spawners such as *Acropora* when coral spawning is disrupted after coral bleaching events (*e.g.*, *Szmant & Gassman, 1990*). This is due to their prolific larval production and monthly reproductive cycle whereas spawning typically only occurs once or twice a year. Similar post-disturbance shifts from *Acropora* to *Pocillopora* dominance have also been documented on several other reefs, including Moorea (*Lenihan et al., 2011*; *Pratchett et al., 2011*; *Adjeroud et al., 2018*) and offshore reefs on the Great Barrier Reef (*AIMS, 2021*). This highlights that live coral cover can show positive recovery trajectories yet the coral community structure may have transitioned to a novel configuration, raising the question of whether this should be considered a 'full' or 'complete' recovery and which metrics should be used in general to assess coral reef recovery (*e.g.*, *Berumen & Pratchett, 2006*).

It is currently unclear over what time scales such post-disturbance shifts from *Acropora* to *Pocillopora* dominance persist and whether they represent a transitional phase indicative of either continuing degradation or recovery (*Aronson et al., 2004*; *Berumen & Pratchett, 2006*). In Moorea, for example, *Pratchett et al. (2011)* argued that the community was unlikely to return to an *Acropora* dominated state because the low number of juveniles present indicated limited recruitment potential. However, a later modelling study predicted substantial recovery of *Acropora* and general reassembly to pre-disturbed levels of coral abundance, composition, and size (*Kayal et al., 2018*). In contrast to Moorea, the percentage of *Acropora* in the subtidal zone at our study site is still high at 40% cover (compared to <1.0% cover in Moorea) (*Pratchett et al., 2011*), thus a return to an *Acropora* dominated state seems possible. Alternatively, *Pocillopora* dominance may even increase further at the expense of *Acropora* corals, or the current co-dominance of *Pocillopora* and *Acropora* may represent an entirely new, stable, and resilient community structure that will endure unless local conditions change or further disturbances occur (*Pratchett et al., 2011*). We caution, however, that our study site represents an environmentally extreme reef environment where lessons from more typical, less extreme reef settings may not necessarily apply.

Following recovery, deficits in functional trait diversity are common on coral reefs as new coral assemblages often have altered species composition that may be deficient in key functional traits, leading to a loss of reef functionality (*Alvarez-Filip et al., 2013*; *McWilliam et al., 2020*). Thus, the post-bleaching shift from *Acropora* to *Pocillopora* dominance in the subtidal likely has implications for the functioning and resilience of this novel coral community. For example, the subtidal coral community also shifted from being dominated by species with a competitive life history strategy (LHS) (*e.g.*, *Acropora* spp.) to being dominated by corals with a weedy LHS (Fig. 6), especially *Pocillopora acuta* which we identified as the most common *Pocillopora* species at our study site (Fig. S2, see Methods). In contrast, the intertidal coral community retained dominance by competitive *Acropora* corals throughout the four survey time points. Such shifts from competitive to weedy and/or stress-tolerant coral species are often observed after disturbances or coral mortality events since weedy corals often survive better and can opportunistically colonize recently disturbed habitats (*Darling, McClanahan & Côté, 2013*; *Kayal et al., 2015*). The weedy LHS is further

characterized by relatively fast reproduction and a brooding reproductive mode (*Darling et al., 2012*), and some species within the *P. damicornis* species complex, such as *P. acuta*, have the ability to produce larvae *via* parthenogenesis (*Stoddart, 1983*; *Schmidt-Roach et al., 2014*). Especially the ability to use parthenogenesis could explain why this genus went from being highly uncommon prior to bleaching (2% cover) to being the dominant coral genus 3.5 years later (44% cover) since this strategy would be particularly advantageous for recruitment at low colony densities in disturbed habitats (*Aronson et al., 2004*; *Darling et al., 2012*; *Carlot et al., 2022*). High photosynthesis rates and resource allocation favoring investment in gamete or larval development over calcification, as observed in *P. verrucosa* from Moorea, may underlie the success of this life history strategy (*Carlot et al., 2022*).

The shift towards *Pocillopora* dominance in the subtidal may also have consequences regarding the resistance to bleaching, storms or outbreaks of coral predators (*e.g.*, *Madin et al., 2014*). For example, this shift was accompanied by a shift from corals with a high corallite integration index score (CIIS) prior to the bleaching event (76% cover) to corals with low CIIS 3.5 years later (55% cover) (Fig. S4). This was primarily due to the shift from dominance by *Acropora* (CIIS of 3.5) to dominance by *Pocillopora* (CIIS of 1.75) (*Swain et al., 2018*). A high CIIS has been linked to a significantly reduced bleaching response because coral colonial integration and coordination improves responses to injury, predation, disease, and stress (*Swain et al., 2018*). This shift to low CIIS dominance could therefore indicate that the subtidal may now be more vulnerable to future bleaching events. However, *Acropora* corals are also highly sensitive to bleaching (*Marshall & Baird, 2000*; *Loya et al., 2001*), thus communities dominated by either *Acropora* or *Pocillopora* may have similar bleaching resistance. Other implications could be a higher resistance of the new subtidal coral community to storms and outbreaks of coral predators such as the crown-of-thorn starfish *Acanthaster planci* because *Acropora* corals have a lower resistance to storm damage than *Pocillopora* (*Berumen & Pratchett, 2006*; *Madin et al., 2014*) and are also the preferred prey of *A. planci* (*Pratchett et al., 2017*).

## CONCLUSIONS

As climate-induced disturbances such as marine heatwaves increasingly impact marine ecosystems globally, understanding how coral reefs respond to, and recover or reassemble after, these disturbances is critical (*Graham, Nash & Kool, 2011*; *Hughes et al., 2018a*; *Tebbett, Morais & Bellwood, 2022*). Although the factors driving recovery trajectories of coral reefs are becoming better understood (*Graham, Nash & Kool, 2011*; *Graham et al., 2015*; *Arif et al., 2022*), it remains unclear whether lessons from more typical coral reefs apply to naturally extreme reef environments, despite their importance as potential refugia and resilience hotspots (*Camp et al., 2018*; *Burt et al., 2020*). The findings from this study demonstrate that spatial patchiness in recovery and coral community reassembly after bleaching might be a common feature on thermally extreme reefs as temperature variability, and therefore coral heat tolerance, can vary dramatically over small spatial scales (hundreds of meters) on such reefs (*Oliver & Palumbi, 2011*; *Schoepf et al., 2015*; *Thomas et al., 2022*). This has important implications for marine spatial planning, coral

reef conservation and management, particularly in the macrotidal Kimberley region where the extreme seascape imposes significant barriers to larval dispersal (*Underwood et al., 2020*). Thermally extreme reef environments, such as those found on shallow reefs in the Kimberley, could play an important role as resilience and adaptation hotspots that may fare better under high-emission climate scenarios than other reef areas (*Schoepf et al., 2015*; *Schoepf et al., 2020*; *Adam et al., 2022*). However, when intensifying climate change is super-imposed on already extreme thermal regimes, the heat tolerance of even naturally heat-resistant coral populations may be overwhelmed in the long-term (*Schoepf et al., 2019*; *Klepac & Barshis, 2020*).

## ACKNOWLEDGEMENTS

We thank: the Bardi Jawi people who enabled this research through their advice and consent to access their traditional lands; J. Brown, G. Firman and the staff at the Kimberley Marine Research Station and Cygnet Bay Pearl Farm for facilitating fieldwork; interns of the Kimberley Marine Research Station for assistance in the field; and Dr. Sebastian Schmidt-Roach for identifying *Pocillopora* species.

### Funding
The authors received no funding for this work.

### Competing Interests
The authors declare there are no competing interests.

### Author Contributions
- P. Elias Speelman analyzed the data, prepared figures and/or tables, authored or reviewed drafts of the article, and approved the final draft.
- Michael Parger performed the experiments, prepared figures and/or tables, authored or reviewed drafts of the article, and approved the final draft.
- Verena Schoepf conceived and designed the experiments, performed the experiments, prepared figures and/or tables, authored or reviewed drafts of the article, and approved the final draft.

### Data Availability
The seawater temperature data, benthic survey data, R code files for the t-test, PCAs and Permanovas are available in the Supplemental Files.

### Supplemental Information
Supplemental information for this article can be found online at http://dx.doi.org/10.7717/peerj.15987#supplemental-information.

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
