# Peer review of "Divergent recovery trajectories of intertidal and subtidal coral communities highlight habitat-specific recovery dynamics following bleaching in an extreme macrotidal reef environment"

_PeerJ, doi:10.7717/peerj.15987_

## Round 0.1 · original submission · Minor Revisions

I have received evaluations of your manuscript from two expert reviewers. Both have made very encouraging comments, and remark on the high quality of your manuscript. Both have made suggestions to improve the manuscript as you can see below. I look forward to receiving your resubmission and rebuttal letter.

Reviewer 1 ·

Basic reporting

In this paper led by Speelman et al. the authors investigate the recovery capacity of intertidal coral communities and compare it to the recovery capacity of subtidal coral communities. To do so, they based their study on the Kimberley region in NW Australia and investigated four time periods: 2016 (January - pre-mass bleaching event, April - mass bleaching event and October - 6 months after the mass bleaching event) and October 2019 (3.5 years after the mass bleaching event). The paper is very well written; the authors have explored and discussed different aspects very well, which made the reading entertaining. The subject is very timely and will be very important in coral conservation research.

Overall comments
I don't have any big criticisms to make; it has been a long time since I reviewed such a good paper. However, I would like to discuss some major points:
- I remain sceptical about the pre-mass bleaching event "coral community health". Besides, it is noticeable in figure 2, the cover was already impacted before the mass bleaching event. Can you discuss this limit?
- I have suggested some studies to add. These are just suggestions, but I think the papers from Speare et al. 2021, Groeneveld & Meeden 1984, Husband et al. 2022, Carlot et al. 2022 and Madin et al. 2014, would be a plus
- I'm a little frustrated with the quality of the figures, which are still very basic. The text is very good, and the science behind it is fascinating, but the figures are relatively basic. Maybe you could change at least the base colours from ggplot?
Again, congratulations, it's a neat paper!

Specific comments
54. This paper might be of interest as well:
Speare et al. 2021 – https://onlinelibrary.wiley.com/doi/full/10.1111/gcb.16000
62. Move today before the parenthesis; it feels awkward.
89. Ease the sentence: "over years"
98. It's amazing! I was not aware of that!
137. You've already mentioned it in the introduction. Maybe don't spoil that much in the introduction
Study site. I feel it misses a map and a picture of these intertidal pools. I'm aware you've added one in the Schoepf et al., 2020 paper, but it would really help. Something like:
A) Kimberley with an inner world map to locate it
B) (intertidal pool ecosystem + subtidal zone) map and
C) true pictures
153. monthly, weekly, daily, hourly…?
166-169. I'm not really a fan of this kind of statement. Every scientific study shows something.
Suggestion: To predict bleaching prevalence, we used the Daily Temperature Range (DTR) because it is one of the most pertinent to highlight such high-frequency temperature variability mitigates bleaching risk (Safaie et al., 2018)
174. I think you need this citation here:
Groeneveld & Meeden 1984 – https://rss.onlinelibrary.wiley.com/doi/abs/10.2307/2987742
174. I would also remove this section: "Although the meta-analysis by Safaie et al. (2018) found these two metrics less influential in predicting bleaching prevalence DTR" and slightly rephrase the end of the sentence to make it fit. It will strengthen your position and give strength to your paper as well!
192. I'd rephrase, "Six 15-meter transects were randomly deployed."
222. You might like to add this citation:
Husband et al. 2022 – https://link.springer.com/article/10.1007/s00338-022-02247-6
244. the first time you need to write down Pocillopora acuta, then you can state it as P. acuta
250. was it not possible to define it? The LHS are broad and should not be complicated to determine.
266. There's nothing to change, but I'm wondering why you've used Excel to run a single t-test which is one single line of code in R and then running PERMANOVA and PERMDISP in R. Or maybe I've misunderstood.
Figure 2. remove substrate; it will improve the readability of the other categories
314. Was there any other pre-disturbance in 2016? I'm wondering because your post disturbances coral cover in intertidal pools was far lower than after. There may be a shift in species assemblages explaining this change, but possibly, communities were impacted before and thus, your observations might be biased. Do you have any documented observations pre-2016?
After looking at Figures 4 and 5, it seems that the change is explained due to Acropora, which fits with this observation. By any chance, do you know which species of Acropora it is? Acropora does not get a high-temperature tolerance, and if it gets a high post-bleaching recovery, it might be more severely affected in pools than in a subtidal environment if repeated/prolonged marine heat waves.
Coral community composition and Life history strategy (LHS) sections. It's really fascinating because this is exactly what you observe in Mo'orea, French Polynesia, for this community change in species (subtidal zone). Adjeroud et al. 2018 talk about it in their papers. Later, Carlot et al. 2022 suggest that Pocillopora might have more energy to colonize bare substrate after a disturbance by not allowing everything in the growth process, such as Acropora. I think you might like to read it.
381-394. Fascinating and really well summarized!
424. my bet would be incomplete yet due to this shift between Pocillopora and Acropora
435. looking at M&M, I haven't seen it written. At what depth was the temperature defined? Have you defined the temperature at the surface or close to the substrate as expected? I'm surprised by the non-change between intertidal and subtidal systems in Figure 1C. Shouldn't it be different? Especially when I see the DTR going up to 7.5°C in the intertidal system. At what time was the weekly temperature defined for the morning or at 12:00 AM?
455-457. Agree with this statement. It also depends on whether there is a shift between species.
486. Excellent!
495. I agree, but it would depend on other disturbances. In addition, you mention the heat stress in this paper, but tabular Acropora might be easily dislodged (see Madin et al, 2014), resulting in a very unlikely colonization succession rate in the future. But hopefully, I might be wrong.
523. I think you need this citation here:
Carlot et al. 2022 – https://onlinelibrary.wiley.com/doi/abs/10.1002/ece3.8613
525. I think you need this citation here:
Madin et al. 2014 – https://onlinelibrary.wiley.com/doi/abs/10.1111/ele.12306
534-536. This is interesting. In general, Acropora would be the first to bleach rather than weedy Pocillopora. However, it is to be confirmed that Acropora gets the lower bleaching resistance.
539. I think you need this citation here:
Madin et al. 2014 – https://onlinelibrary.wiley.com/doi/abs/10.1111/ele.12306

Experimental design

Everything'fine here.
Need to look at this point mentioned in the basic reporting:

- I remain sceptical about the pre-mass bleaching event "coral community health". Besides, it is noticeable in figure 2, the cover was already impacted before the mass bleaching event. Can you discuss this limit?

Validity of the findings

Very timely and extremely interesting!

Additional comments

/

·

Basic reporting

In my opinion this is a fantastic paper. I rarely see papers that have almost 0 formatting/language/spelling errors, or papers that are so well written. This paper checks all the “nit-picky reviewer” boxes and shows just how much care the authors have put into the manuscript. On content, the authors describe an important ecological change in reefs in a unique area of Australia following regional bleaching events. They report that bleaching led to ecological change in subtidal habitats that were not reflected in the intertidal habitats, notably that subtidal reefs are becoming more weedy, whereas intertidal reefs show signs of resilience and stress tolerance. I think this study fits nicely within the body of work from the Kimberly region and highlights the ecological changes that are occurring (as well as hypotheses to explain them) as a result of thermal bleaching. The study is sound with appropriate data collection, reporting, and statistical approaches. In total, my comments represent minor suggestions. This may be my most succinct review to date – a testament to the author’s hard work. Nicely done.

The authors did a very nice job at giving a background for their study location, provided robust data to contextualize the unique attributes of these reefs, and provided a through well-written and sourced discussion. I did not see a place where the raw data and code would be deposited, so please make sure this is clear in the MS.

Experimental design

The design of the study was appropriate and statistical analyses fitting for the questions and contrasts that were tested. I had a question regarding transformations in the PERMANOVA and PCA, but in my experience with these approaches in R, I think the author’s likely have the correct “scaling/rotating/centering”, although some detail to this point would be helpful in the MS. Might it be useful to

Validity of the findings

Findings are valid and well

Additional comments

Overall the Intro is masterfully written. A very nice, thorough (yet concise) read, with hypotheses and objectives clearly detailed at the end of the section. Well done!
Line 58: maybe mention cytotoxic stress here, not just a nutritional issue
Line 63: here and elsewhere make sure to give a line return with a space or an indent at new paragraphs

Methods
Line 160: maybe put “ “ around w > MMM at first mention?
Line 170: delete extra space after 2019
Line 268: not to be “R-snobby” but can you do the t-test in a program designed to do stats vs. excel? You do so much more complicated work here that the “t-test in excel” seems odd.
Line 272: Permanova with data at different scales usually has to be “scaled and centered” or something similar – this is usually done as a default, but can you confirm if this was done? I’m not clear if your data are all at different scales (like it would be for different physiological metrics), so perhaps your distribution is still 0-1 and/or categorical? Just a thought.
Line 287: data and code available online? Github? Zenodo? Dryad?
Line 332: italics for Porites
Line 335 and 339: looks like double space before the (Fig….)
Table 1. I think this could be set at 3 decimals instead of 3, 4, and 5 in different columns
Figure 3 and 5c. I think the ggplot “theme_bw()” or “theme_classic” could be applied here to clean up the figure and remove the gray background.

---

## Round 0.2 · accepted · Accept

I am satisfied with the minor modifications that have been made to the manuscript. Congratulations!